# IL-10 in Systemic Lupus Erythematosus: Balancing Immunoregulation and Autoimmunity

**DOI:** 10.3390/ijms26073290

**Published:** 2025-04-02

**Authors:** Patricia Richter, Luana Andreea Macovei, Ciprian Rezus, Vasile Lucian Boiculese, Oana Nicoleta Buliga-Finis, Elena Rezus

**Affiliations:** 1Department of Rheumatology and Rehabilitation, “Grigore T. Popa” University of Medicine and Pharmacy, 700115 Iasi, Romania; patricia.richter@umfiasi.ro (P.R.); elena.rezus@umfiasi.ro (E.R.); 2I Rheumatology Clinic, Clinical Rehabilitation Hospital, 14 Pantelimon Halipa Street, 700661 Iasi, Romania; 3Department of Internal Medicine, “Grigore T. Popa” University of Medicine and Pharmacy, 16 University Street, 700115 Iasi, Romania; oana-nicoleta.buliga-finis@umfiasi.ro; 4III Internal Medicine Clinic, “St. Spiridon” County Emergency Clinical Hospital, 1 Independence Boulevard, 700111 Iasi, Romania; 5Department of Medical Informatics and Biostatistics, “Grigore T. Popa” University of Medicine and Pharmacy, 700115 Iasi, Romania; vasile.boiculese@umfiasi.ro

**Keywords:** systemic lupus erythematosus, interleukin-10, interleukin-6

## Abstract

Interleukin-10 (IL-10) presents a dual role in systemic lupus erythematosus (SLE), illustrating pro-inflammatory and anti-inflammatory effects. This study addressed the possible associations between serum IL-10 levels and demographic characteristics, laboratory parameters, organ manifestations, disease activity, and treatment response in SLE patients. A total of 88 SLE patients from the Rheumatology Clinic of the Clinical Rehabilitation Hospital, Iași, were enrolled. Disease activity was assessed using the Systemic Lupus Erythematosus Disease Activity Index (SLEDAI). Serum cytokine levels were measured using an enzyme-linked immunosorbent assay (ELISA). Serum IL-10 levels were significantly higher in males compared to females (47.62 pg/mL vs. 13.24 pg/mL, *p* = 0.011) but not significantly associated with age or disease duration. However, IL-10 showed positive correlations with inflammatory markers and autoantibodies, including C-reactive protein (*p* = 0.002), IL-6 (*p* = 0.01), ANA (*p* = 0.014), and anti-SSB antibodies (*p* = 0.005). Our findings indicate that IL-10 may be involved in inflammatory and immune processes in SLE, as evidenced by its significant correlations with specific autoantibodies and inflammatory markers in our study. However, IL-10 did not correlate with disease activity, organ involvement, or treatment response. These results underline the participation of IL-10 in SLE and emphasize the need for further research to clarify its potential as a biomarker or therapeutic target.

## 1. Introduction

Systemic lupus erythematosus (SLE) represents a complex and multifaceted autoimmune disease that affects several organs and has varying clinical consequences [1].

The etiology of SLE, though not fully understood, is attributed to a succession of genetic predilection, hormonal influences, and environmental exposures leading to immune dysfunctions. A distinctive feature of SLE is the production of autoantibodies by T-helper (Th) cells, which causes immune complex formation and subsequent tissue damage. This process can clinically manifest in severe forms, including hematological, neuropsychiatric, and cardiovascular complications, while milder forms primarily affect the skin and joints. Immune system failure and persistent inflammation are the underlying causes of these manifestations [2,3,4,5].

In addition to the well-known genetic or environmental contributors, an imbalance between pro-inflammatory and anti-inflammatory cytokines illustrates immune dysregulation, which initiates inflammatory responses and causes damage to tissues and organs. In SLE, these inflammatory processes can be regulated by excessively produced cytokines. As soluble mediators, cytokines are required to coordinate and regulate immune cells’ differentiation, maturation, and activation through various interconnected pathways. Not only do they mediate chronic inflammation, but they also disrupt the physiological balance essential for maintaining tissue integrity, thereby facilitating the progression of SLE [6,7,8,9].

Monitoring and evaluating prognosis in clinical practice routinely include measuring specific cytokines along with other serological markers, like anti-double-stranded DNA (anti-dsDNA) and complement levels, to assess disease activity and therapeutic response [6,10]. Cytokines, proteins secreted by various cell types, attach to receptors on the surface of cells to exert different effects on cellular signaling and communication [11]. Because their involvement in SLE is significant, they can potentially be used as biomarkers to monitor disease progression. A more profound comprehension of cytokine functions and interactions could help identify clinical SLE phenotypes and enhance the management approach [6,12,13].

Pro-inflammatory cytokines such as interleukin-1 (IL-1), IL-6, tumor necrosis factor-alpha (TNF-α), and interferons (IFNs) may worsen inflammation and contribute to SLE symptoms, and they are considered indicators of disease activity [6,14]. Conversely, several anti-inflammatory cytokines, including those from the IL-1 and IL-10 families, can down-regulate inflammatory responses by influencing immune activation and promoting self-tolerance [7]. Immunoregulatory cytokines, such as IL-10 and transforming growth factor (TGF)-β, are present at elevated levels in SLE, contributing to its pathogenesis [7,15].

Given IL-10’s dual immunoregulatory and pro-inflammatory functions [7], we aimed to investigate its clinical relevance in SLE. Despite growing interest in IL-10 as a biomarker, existing studies offer conflicting results regarding its correlation with disease activity and clinical features, and few address these associations in real-life, treatment-exposed cohorts. In this original study, we analyzed the correlation between serum IL-10 levels and specific autoantibodies, disease activity, and clinical manifestations including arthritis, serositis, skin involvement, renal, hematological, and neurological involvement in a cohort of SLE patients.

## 2. Results

### 2.1. Demographic and Clinical Characteristics

The 88 patients with SLE enrolled in the study had a mean disease duration of 10.38 ± 9.73 years, varying between 0 and 37 years (Table 1). The study population consisted primarily of female patients, with 79 female and 9 male participants, resulting in a female-to-male ratio of approximately 9:1. The mean age of the cohort was 51.17 ± 15.37 years, with a median of 54 years. Patient ages ranged from 18 to 77 years, with an interquartile range (IQR) of 23 years, indicating moderate variability in age distribution.

### 2.2. IL-10 Levels and Gender

In the cohort, mean IL-10 levels were higher in males (47.62 ± 73.91 pg/mL) than in females (13.24 ± 11.33 pg/mL), as illustrated in Figure 1. The minimum and maximum values ranged from 8.1 to 236.8 pg/mL in males and 6.4 to 95.4 pg/mL in females. The median IL-10 levels were 16.09 pg/mL in males (IQR: 44.6 pg/mL) and 10.41 pg/mL in females (IQR: 4.4 pg/mL). Notably, only one male patient had an IL-10 level exceeding 200 pg/mL (236.8 pg/mL), and this value contributed to the overall dispersion of IL-10 values within the male subgroup. The Mann–Whitney U test further supported this trend, with a mean rank of 65.00 in males and 42.16 in females, suggesting higher IL-10 levels in the male subgroup. A notable correlation between gender and IL-10 levels was confirmed, which was statistically significant (*p* = 0.011).

### 2.3. IL-10 Levels and Participants’ Age

Spearman’s and Pearson’s correlation analyses were performed to analyze the interplay between IL-10 levels and patient age. Both correlation coefficients indicate no significant association between IL-10 levels and age (Spearman’s r = 0.000, *p* = 0.999; Pearson’s r = −0.104, *p* = 0.333).

### 2.4. IL-10 Levels and Disease Duration

Pearson (r = −0.070, *p* = 0.517) and Spearman (r = −0.048, *p* = 0.655) correlation analyses indicate no significant association between IL-10 levels and disease duration.

### 2.5. IL-10 and Disease Activity

The mean Systemic Lupus Erythematosus Disease Activity Index (SLEDAI) score in our cohort was 3.61 (range: 0–12). Of the 88 patients, 22 had a clinically significant SLEDAI score ≥ 6, indicating active disease, while the majority (66 patients) had a SLEDAI score of <6, reflecting a predominantly inactive disease state.

Although the mean IL-10 levels were slightly higher in the active disease group (18.5 ± 19.7 pg/mL) compared to the inactive group (16.2 ± 29.1 pg/mL), the median values were nearly identical (11.1 vs. 10.5 pg/mL). The Mann–Whitney U test (mean rank: 52.23 vs. 41.92) suggested a trend toward higher IL-10 levels in active disease, yet this difference did not reach statistical significance (*p* = 0.101). Moreover, when assessed through linear regression, the Pearson correlation coefficient indicated a weak inverse association (r = −0.028), with no statistical significance (*p* = 0.398). The ANOVA analysis further confirmed the lack of a significant relationship between IL-10 and the SLEDAI score (*p* = 0.795), as illustrated in Figure 2.

### 2.6. IL-10 Levels and Disease Manifestations

In this SLE cohort (N = 88), immunological (97.7%), articular (79.5%), and hematological (78.4%) manifestations were most common, followed by cutaneous (65.9%) and renal involvement (31.8%).

In this cohort, IL-10 levels were not significantly associated with most clinical manifestations of SLE Table 2. While patients with neurological involvement exhibited a trend toward higher IL-10 levels (22.0 pg/mL vs. 15.2 pg/mL, *p* = 0.085), this finding did not reach statistical significance. Similarly, renal (19.6 pg/mL vs. 15.4 pg/mL, *p* = 0.237), hematologic (17.8 pg/mL vs. 12.9 pg/mL, *p* = 0.659), mucosal (29.4 pg/mL vs. 14.6 pg/mL, *p* = 0.920), and cardiovascular involvement (21.4 pg/mL vs. 16.6 pg/mL, *p* = 0.186) showed slightly elevated IL-10 levels, but these differences were not statistically conclusive.

However, when patients were stratified based on renal involvement, mean IL-10 levels were higher in those with renal manifestations (31.10 ± 42.90 pg/mL) compared to those without renal involvement (16.07 ± 26.18 pg/mL).

### 2.7. IL-10 Levels and Laboratory Markers

The mean IL-10 level in the cohort was 16.75 ± 26.96 pg/mL, with values ranging from 6.44 to 235.81 pg/mL. Based on the manufacturer’s cutoff for IL-10 (<10.8 pg/mL), 45.5% of patients had IL-10 levels ≥ 10.8 pg/mL, while 54.5% had levels below this threshold. One notable finding is the positive correlation between IL-10 and C-reactive protein (CRP) (r = 0.327, *p* = 0.002) (Figure 3). Neutrophils (r = 0.197, *p* = 0.066) and erythrocyte sedimentation rate (ESR) (r = 0.191, *p* = 0.075) showed trends toward positive correlations, but these did not reach statistical significance. No significant correlations exist between IL-10 and other hematological parameters, such as hemoglobin, hematocrit, lymphocytes, or platelets.

Serum IL-10 levels were higher in patients with an active inflammatory syndrome (elevated ESR and/or CRP) (18.71 ± 18.25 pg/mL) compared to those without (15.27 ± 32.17 pg/mL). However, this difference did not reach statistical significance.

IL-10 showed a significant positive correlation with alkaline phosphatase (r = 0.228, *p* = 0.033). Other parameters, including urea, creatinine, glucose, uric acid, liver enzymes, and lipid profile, presented weak or no correlations with IL-10 levels, with no significant findings; the detailed results are presented in Table 3.

In the immunological profile, IL-10 levels were significantly correlated with antinuclear antibody (ANA) positivity (r = 0.274, *p* = 0.014) and anti-SS-B antibodies (r = 0.310, *p* = 0.005). A trend toward significance was observed for anti-SS-A antibodies (r = 0.185, *p* = 0.098). No significant correlations were observed between IL-10 and anti-dsDNA antibodies, antiphospholipid antibodies, or complement levels.

A statistically significant positive correlation was observed between IL-10 and IL-6 (r = 0.274, *p* = 0.01). However, correlations between IL-10 and IL-17 (r = 0.144, *p* = 0.181) and between IL-10 and TNF-α (r = 0.165, *p* = 0.125) did not reach statistical significance, indicating no strong association in this cohort.

Moreover, our patients were divided into two subgroups based on the presence of anti-dsDNA antibodies. Among the 88 SLE patients, 45.5% were positive for anti-dsDNA antibodies. In this subgroup, IL-10 correlated positively with anti-dsDNA (r = 0.332, *p* = 0.036) and RNP 70 (r = 0.486, *p* = 0.012), while no significant associations were observed with SS-A or SS-B antibodies. In contrast, in anti-dsDNA negative patients, IL-10 was significantly associated with CRP (r = 0.314, *p* = 0.030), ALP (r = 0.367, *p* = 0.010), ANA (r = 0.328, *p* = 0.036), and anti-SSB (r = 0.406, *p* = 0.006), while showing a negative correlation with albumin (r = −0.302, *p* = 0.037). Additionally, IL-10 correlated with IL-6 (r = 0.323, *p* = 0.025) in anti-dsDNA-negative patients.

### 2.8. IL-10 Levels and Treatment

Steroids were used by 39.8% of patients, while NSAIDs were prescribed to 13.6%. Hydroxychloroquine remained the most common treatment (89.8%), consistent with its role as a cornerstone treatment in SLE. Among immunosuppressants, azathioprine (33.0%) was the most frequently used, followed by methotrexate (9.1%) and mycophenolate mofetil (5.7%). Belimumab was administered to 8.0% of patients. However, IL-10 levels showed no significant association with any of these treatments. No differences were observed for NSAIDs (*p* = 0.363) or corticosteroids (*p* > 0.850). While IL-10 levels were lower in methotrexate users (12.30 vs. 17.14 pg/mL, *p* = 0.740), this difference was insignificant. Similarly, no significant differences were found in patients receiving hydroxychloroquine, azathioprine, MMF, cyclophosphamide, or belimumab.

## 3. Discussion

In this study, we evaluated serum IL-10 levels in a real-life cohort of SLE patients and analyzed their association with disease activity, clinical manifestations, and immunological parameters. As expected, the study population consisted predominantly of female patients, reflecting the well-established higher prevalence of SLE in women. Interestingly, IL-10 levels were significantly higher in male patients compared to females; however, this observation was influenced by an extreme value within the small male subgroup. No correlations were found between IL-10 levels and disease activity, specific organ involvement, or current treatment regimens. In contrast, significant positive correlations were identified between IL-10 and IL-6, ANA, and CRP levels.

SLE is characterized by a complex interplay of immunological factors, where dysregulation in the immune system triggers an abnormal elevation of various cytokines [7,8,16]. Key players include type I and II interferons, IL-6, TNF-α, and immunomodulatory cytokines such as IL-10 and TGF-β [17]. Th2 cytokines, particularly IL-10, are frequently implicated in the development of SLE [3]. Both adaptive and innate immunity cells, such as dendritic cells and macrophages, mast cells, killer cells, eosinophils, neutrophils, B cells, CD8+ T cells, and distinct T-helper subsets such as Th1, Th2, Th17, and regulatory T cells, are known to produce IL-10 [11]. Research increasingly supports the relevance of IL-10 in autoimmune conditions, suggesting its involvement in the regulatory mechanisms underlying these conditions [18]. Despite its generally recognized anti-inflammatory properties, the function of IL-10 in SLE remains controversial. Some reports affirm that the cytokine is positively associated with disease activity and damage index, while others do not support this relationship [6,19,20,21]. IL-10 is thought to be protective against autoimmune diseases by inhibiting pathogenic inflammation and promoting self-tolerance. It reduces inflammation by decreasing the production of pro-inflammatory cytokines, reducing the antigen-presenting capacity of cells and thus suppressing Th cell functions [6]. A considerable number of human investigations have revealed elevated levels of IL-10 in the serum of SLE patients when compared to controls [3,18,20,22,23,24,25,26,27,28,29,30,31,32,33,34,35,36,37,38,39,40]. Consistent with previous studies [23,36], our analysis found no correlation between IL-10 levels and patient age or disease duration, reinforcing the lack of a clear age-dependent pattern in IL-10 levels.

Sex-based differences in immune function have been extensively documented. Women have been demonstrated to respond more protectively than men to immunologically stimulating events. The high female predominance in autoimmune diseases supports the notion that women, while having stronger immune responses, may also be more susceptible to developing autoimmunity and losing self-tolerance [41]. In this context, sex hormones appear to influence IL-10 production through multiple mechanisms. Estrogen regulates IL-10 expression in peripheral blood mononuclear cells from SLE patients, while androgens such as testosterone enhance Th2 differentiation and IL-10 production by T cells and also act directly on CD4^+^ T lymphocytes to increase IL-10 levels [42,43]. Interestingly, in contrast, progesterone and estradiol have been reported to inhibit IL-10 production by activated marginal zone B cells, suggesting a complex, context-dependent regulation of this cytokine [44]. In light of these observations from the literature, we also explored the relationship between gender and IL-10 levels in our cohort.

Notably, our analysis revealed a significant correlation between IL-10 levels and gender, with higher IL-10 concentrations in male patients compared to females. However, the small number of male participants limits the strength of this observation. Therefore, the potential influence of sex hormones on IL-10 levels in SLE should be further investigated in larger and sex-balanced cohorts.

While our study did not detect a link between IL-10 levels and clinical manifestations or disease activity as assessed by the SLEDAI score, others have suggested that IL-10 release fluctuates according to the disease phase, further reflecting its dual role in SLE progression. The high variability in IL-10 levels, indicated by the wide standard deviation, may have contributed to the lack of significant association between IL-10 levels and SLEDAI scores. Numerous studies have consistently found a direct relationship between IL-10 levels and disease activity, most frequently measured by the SLEDAI score. This indicates that IL-10 values tend to increase in parallel with an increase in the SLEDAI score, which quantifies the severity of the disease [3,18,20,27,28,29,30,31,32,38,45]. Two longitudinal studies confirmed the relationship, providing valuable insights into the disease’s progression by observing trends and changes in IL-10 levels and their correlation with SLE disease activity over time [29,46]. These findings suggest that IL-10 could be considered a biological marker of disease activity in SLE patients.

This perspective is also shared by Alhassbalawi et al. who showed that newly diagnosed SLE patients or those experiencing a flare may present a more pronounced increase in IL-10 levels than patients in remission or those undergoing treatment. Their findings demonstrated notable differences in IL-10 expression between new-onset and treated SLE patients, suggesting that elevated IL-10 levels in recently diagnosed individuals may reflect an active immune response or flare-up [47]. Longitudinal studies on lupus nephritis patients have shown a significant decrease in IL-10 levels from the time of inclusion to remission; at remission, IL-10 concentrations became comparable to those observed in SLE patients with inactive disease [48].

Conversely, other research obtained no clear association between IL-10 and active disease [49], while Arora et al. even identified a negative relationship between IL-10 levels and SLEDAI scores [33]. This heterogeneity across studies may be explained by differences in patient populations, treatment regimens, disease stage, or the timing of cytokine measurement. In our cohort, the absence of a significant correlation between IL-10 and disease activity may be partly attributed to the generally low SLEDAI scores, which likely reflect the effect of background immunosuppressive therapy.The connection between IL-10 and disease activity is further evidenced by its correlations with laboratory markers [46]. Elevated anti-dsDNA titers in SLE patients may be related to the IL-10’s potential to promote B cell growth, differentiation, proliferation, and antibody production while also preventing the apoptosis of auto-reactive B-cells [46]. Several studies have reported a positive correlation between serum IL-10 levels and anti-dsDNA antibody titers, indicating its involvement in autoantibody production [30,32,45,46]. In contrast, studies by Winikajtis-Burzyńska et al. and Park et al. did not identify a significant correlation between IL-10 and anti-dsDNA titers [23,29]. Consistent with these findings, our study did not observe an evident relationship between IL-10 and anti-dsDNA levels. Notably, in our study, IL-10 correlated positively with anti-dsDNA antibody titers but only within the subgroup of anti-dsDNA positive patients, suggesting a potentially stronger association in selected patient populations.

Previous studies have highlighted significant associations between IL-10 and hematological parameters, including a negative correlation with hemoglobin and lymphocyte levels [23,32,50,51]. However, in our study, IL-10 presented no association with these indices.

In addition a positive association between IL-10 and inflammatory markers, including ESR, CRP, and fibrinogen levels, was established [23]. In line with these results, our study also described a significant correlation between serum IL-10 levels and CRP, a relationship previously documented by Elazeem et al. [36] and Winikajtis-Burzyńska et al. [23]. However, conflicting data exists; Lacki et al. identified no specific correlation between SLE IL-10 levels and acute phase proteins [28].

Complement components C3 and C4 are typically associated with disease activity in SLE, frequently decreasing in active disease due to complement consumption and immune complex formation [52]. Both C3 [26,32,45,46,53] and C4 levels [25] have been reported to correlate significantly with IL-10, indicating a possible relationship between complement activation and IL-10-mediated immune modulation. However, our study did not identify a significant association between IL-10 and complement fractions. Furthermore, findings in the literature remain inconsistent, as Park et al. also found no correlation between IL-10 and complement levels [29].

Studies have explored the association between IL-10 and autoantibody production, particularly in the case of anti-SSA and anti-SSB antibodies. High serum IL-10 levels (>5.11 pg/mL) have been linked to an increased risk of anti-SSA positivity [23]: IL-10 has been correlated with anti-Sm, anti-SSA, and anti-SSB antibody titers [54]. Interestingly, our study also found a significant correlation between IL-10 levels and anti-SSB, confirming its potential function in autoantibody regulation in SLE.

Different cytokines have been investigated in connection with IL-10, offering valuable data on its place in immune-mediated dysregulation in SLE. A previous study has shown a meaningful correlation between IL-10 and IL-6, suggesting a shared involvement in SLE-related inflammation [23]. Similarly, Lacki et al. [32] and Waszczykowska et al. [27] found a significant association between these cytokines. In another report, no significant relation was seen between IL-10 and IL-17 [2]. Our study aligns with these observations, as we also identified a significant correlation between IL-10 and IL-6, whereas IL-10 displayed no evident association with IL-17. However, previous research has reported a notable correlation between IL-10 and TNF-α [27]. Our study did not confirm this association.

As a final reflection, we provided a synthesis of our results. To highlight the role of this cytokine, we interpreted our findings in relation to previously published studies on IL-10 in SLE. We did not detect a significant correlation between IL-10 levels and disease activity as measured by the SLEDAI; this may reflect the predominantly inactive disease state in our cohort. While previous studies have reported a negative correlation between IL-10 levels and hematological parameters such as hemoglobin and lymphocyte counts [23,32,50,51], our study did not confirm these associations. Similarly, no relationship was observed between IL-10 levels and complement components [25,26,32,45,46,53]. On the other hand, our results align with studies reporting a positive association between IL-10 and inflammatory markers such as CRP [23,36], supporting IL-10’s involvement in systemic inflammation. A significant correlation was also found between IL-10 and anti-SSB antibodies, consistent with earlier research highlighting IL-10’s link to humoral autoimmunity [54]. Contrary to previous studies reporting a general correlation between IL-10 and anti-dsDNA antibodies [3,30,32,45,46], our findings showed this association only within the anti-dsDNA positive subgroup. Moreover, our study confirmed a strong positive relationship between IL-10 and IL-6, in agreement with previous findings, whereas no association was found with IL-17 or TNF-α [2,27,28]. Regarding clinical manifestations, although data from the literature reported a correlation between IL-10 and neurological involvement [34], IL-10 levels were not significantly associated with most clinical features in our cohort.

Although our findings add to the current understanding of IL-10 in SLE, some limitations should be acknowledged. Serum IL-10 was measured only once per patient, which did not allow us to assess intra-individual variability or fluctuations over time. The number of male participants was low (N = 9), limiting the statistical power to explore sex-related differences in IL-10 levels. Moreover, one male patient had an extremely elevated IL-10 value, which may have disproportionately influenced group-level comparisons. The study did not include a healthy control group, which limits the ability to determine whether IL-10 levels were elevated compared to baseline physiological levels. Future research should focus on larger, prospective, and sex-stratified cohorts to further elucidate the role of IL-10 in disease pathogenesis, activity, and as a potential biomarker in SLE.

## 4. Materials and Methods

### 4.1. Study Group

A total of 88 SLE patients treated at the Rheumatology Clinic of the Clinical Rehabilitation Hospital Iași, from July to November 2022 were included. All patients fulfilled the classification criteria for SLE based on either the 1997 revised American College of Rheumatology (ACR) criteria [55] or the 2012 Systemic Lupus International Collaborating Clinics (SLICC) classification criteria [56] at the time of diagnosis. Patients with overlap syndromes, simultaneous SARS-CoV-2 infection, unwillingness to participate, or cognitive and communication disorders were excluded.

Our study was guided by the principles outlined in the Declaration of Helsinki. Prior approval was obtained from the Clinical Rehabilitation Hospital Ethics Committee, Iași, and the “Grigore T. Popa” University of Medicine and Pharmacy, Iași. All patients provided informed consent.

### 4.2. Clinical Assessment

Patients’ date of birth, gender, disease activity, organ involvement, and current treatment were retrieved from their medical records. SLEDAI score was utilized to assess disease activity during consultation. Using 24 clinical and laboratory indicators, the SLEDAI expresses disease activity over the last 30 days [57,58]. A SLEDAI score ≥ 6 was considered indicative of active disease.

### 4.3. Laboratory Assessment

Routine laboratory assessments included the hemoglobin levels; leukocyte, lymphocyte, and platelet counts; ESR using the Westergren method; CRP levels determined by turbidimetric nephelometry; and complement factors C3 and C4 measured by nephelometry. Additionally, hepatic and renal biochemistry parameters were measured. The results were interpreted based on the hospital’s established and validated reference ranges.

Anti-dsDNA antibodies were declared positive if they exceeded 25 U/mL. ANAlevels were measured using an ELISA, and results were expressed as index values. According to the manufacturer’s interpretation guide, values < 1.0 were considered negative, 1.0–1.2 were borderline, and >1.2 were positive. Hypocomplementemia was defined as C3 and/or C4 fractions below the laboratory normal range (C3: 88–252 mg/dL, C4: 13–75 mg/dL). Anti-SSA and anti-SSB positivity were defined as levels > 25 U/mL.

Serum IL-10 concentrations were measured once for each patient at the time of enrollment using a commercially available enzyme-linked immunosorbent assay (ELISA) kit (BioVendor, RD194572200R), following the manufacturer’s instructions. This sandwich immunoassay employs monoclonal anti-human IL-10 antibodies for detection.

Blood samples were collected using standard venipuncture techniques, and serum was separated by centrifugation. Aliquots were stored at −80 °C until further analysis to prevent cytokine degradation. Before the assay, samples were thawed, mixed (vortexed), and diluted threefold (1:3) with the provided Dilution Buffer to optimize detection within the assay’s dynamic range.

Assay procedure: Standards, blanks, and diluted serum samples were pipetted into pre-coated microtiter wells and then incubated for 60 min at room temperature with continuous shaking (300 rpm). After a washing step, a biotinylated anti-IL-10 antibody was added, followed by a second incubation for 1 h. Wells were rewashed before adding Streptavidin–HRP conjugate, which was incubated for another 30 min. After an additional washing step, 100 µL of substrate solution was added, and the resulting solution was left to stand at room temperature for incubation for 10 min.

An acidic stop solution was added to interrupt the reaction, and a microplate reader was used to measure absorbance at 450 nm.

Sensitivity of IL-10 measurement and reference values: A standard curve was generated using recombinant IL-10 standards ranging from 3.13 to 200 pg/mL, with unknown concentrations interpolated accordingly. The assay had a lower detection limit of 1.32 pg/mL. In a reference population of healthy individuals, IL-10 levels were below 3.13 pg/mL. The results obtained from patient samples were multiplied by the dilution factor (×3) to determine the final concentration.

### 4.4. Statistical Analysis

For the descriptive analysis, quantitative variables were stated as minimum, maximum, and mean ± standard deviation (SD) or as median with an interquartile range (IQR). In contrast, qualitative variables were presented as frequencies/percentages.

Considering that our data did not follow a parametric distribution, Spearman’s rank correlation test was used to analyze correlations, including those between serum IL-10, serum IL-6, IL-17A, TNF-alpha, and laboratory markers of disease activity, with corresponding R values calculated; the Mann–Whitney U test was applied to assess differences between groups.

IBM SPSS version 28.0 was used to conduct statistical analyses and create graphs. A significant value of *p* < 0.05 was established for all analyses.

## 5. Conclusions

The impact of IL-10 in SLE appears to be dual and paradoxical. While it promotes to B cell activation and antibody production, contributing to disease exacerbation, it also presents immunoregulatory properties that may offer protection against severe manifestations. Therefore, its clinical interpretation should be integrated with other disease activity measures. In our study, IL-10 correlated significantly with IL-6, ANA, and CRP levels, suggesting a link with systemic inflammation and humoral autoimmunity. However, IL-10 levels showed no association with disease activity scores, specific clinical manifestations, hematological and complement parameters, or treatment regimens. These findings reinforce the idea that IL-10 may reflect immune activation rather than organ-specific damage or clinical activity. Given its contradictory effects, therapies targeting IL-10 must balance immune suppression and regulation. Further research is needed to clarify its role in disease progression and refine targeted interventions.

## Figures and Tables

**Figure 1 ijms-26-03290-f001:**
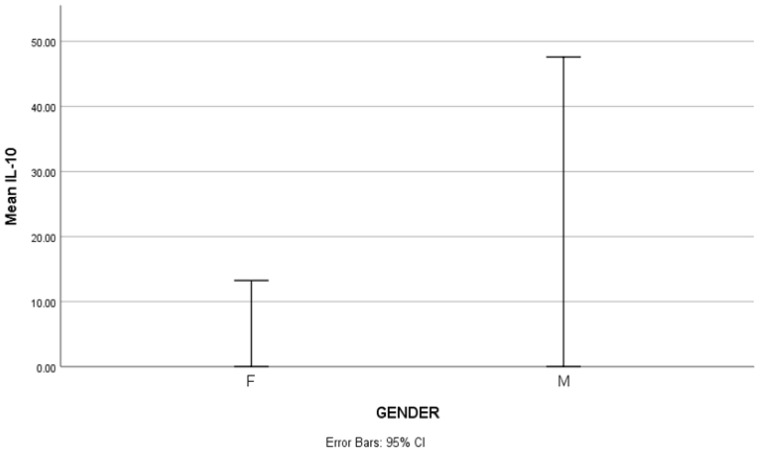
Graph showing that IL-10 levels were higher in male (M) patients compared to females (F); the difference was statistically significant (*p* = 0.011).

**Figure 2 ijms-26-03290-f002:**
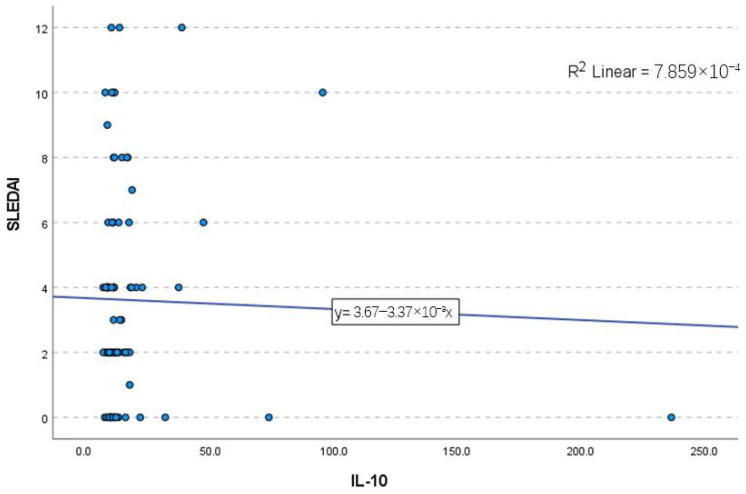
Scatter plot visualizing the lack of association between IL-10 levels and SLEDAI score.

**Figure 3 ijms-26-03290-f003:**
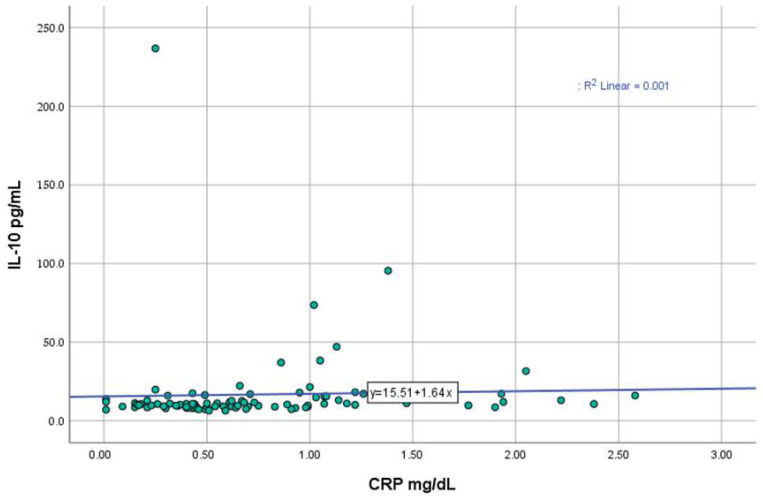
Scatter plot illustrating the relationship between IL-10 and CRP levels.

**Table 1 ijms-26-03290-t001:** Association of IL-10 levels with demographic and clinical characteristics.

Parameter	Mean ± SD	*p*-Value (Pearson)
Disease Duration (years)	10.38 ± 9.73	0.517
Age (years)	51.17 ± 15.37	0.333
SLEDAI Score	3.61 ± 3.24	0.398

**Table 2 ijms-26-03290-t002:** IL-10 levels across different SLE manifestations.

Disease Manifestation	Percentage (%)	*p*-Value (Mann–Whitney U Test)
Articular Involvement	79.5%	0.357
Cutaneous Involvement	65.9%	0.874
Mucosal Involvement	14.8%	0.920
Hematologic Involvement	78.4%	0.659
Immunologic Involvement	97.7%	0.576
Serositis	17.0%	0.951
Renal Involvement	31.8%	0.237
Neuropsychiatric Involvement	22.7%	0.085
Cardiovascular Involvement	3.4%	0.186
Pulmonary Involvement	5.7%	0.987

**Table 3 ijms-26-03290-t003:** Correlations between IL-10 levels and hematological, biochemical, and immunological markers.

Assessed Parameter	Spearman’s Rank Correlation Coefficient (R)	*p*-Value
Hematological Parameters		
RBCs	−0.071	0.51
HGB	−0.07	0.518
HCT	−0.013	0.902
PLTs	−0.009	0.935
WBCs	0.146	0.176
NEUs	0.197	0.066
LYMPHs	−0.113	0.297
Biochemical Parameters		
Urea	0.089	0.411
CREATE	0.151	0.161
UA	0.076	0.482
GLU	0.175	0.102
AST	0.031	0.773
ALT	−0.01	0.927
GGT	−0.12	0.265
ALP	0.228	0.033
LDH	0.023	0.832
TC	−0.099	0.357
TG	−0.06	0.584
CK	−0.107	0.327
ESR	0.191	0.075
CRP	0.327	0.002
Immunological Parameters		
ANA	0.274	0.014
Anti-dsDNA	0.163	0.13
SSA	0.185	0.098
SSB	0.31	0.005
RNP 70	0.243	0.080
aCL IgM	−0.267	0.377
aCL IgG	0.165	0.573
aβ2-GP I IgG	−0.119	0.713
aβ2-GP I IgM	−0.243	0.498
C3	0.001	0.994
C4	−0.008	0.941
Proinflammatory cytokines		
IL-6	0.274	0.010
IL-17A	0.144	0.181
TNF-α	0.165	0.125

RBCs = Red Blood Cells; HGB = Hemoglobin; HCT = Hematocrit; PLTs = Platelets; WBCs = White Blood Cells; NEUs = Neutrophils; LYMPHs = Lymphocytes; CREA = Creatinine; UA = Uric Acid; GLU = Glucose; AST = Aspartate Aminotransferase; ALT = Alanine Aminotransferase; GGT = Gamma-Glutamyl Transferase; ALP = Alkaline Phosphatase; LDH = Lactate Dehydrogenase; TC = Total Cholesterol; TG = Triglycerides; CK = Creatine Kinase; ESR = Erythrocyte Sedimentation Rate; CRP = C-Reactive Protein; ANA = Antinuclear Antibodies; Anti-dsDNA = Anti-Double-Stranded DNA Antibodies; SSA = Anti-SSA Antibodies; SSB = Anti-SSB Antibodies; aCL = Anticardiolipin Antibodies; aβ2-GP I = Anti-β2-Glycoprotein I; aPL = Antiphospholipid Antibodies; C3 = Complement C3; C4 = Complement C4; IL-6 = Interleukin-6; IL-17A = Interleukin-17A; TNF-α = Tumor Necrosis Factor Alpha.

## Data Availability

The original contributions presented in this study are included in the article. Further inquiries can be directed to the corresponding authors.

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
