# Peer review of "IL-10 in Systemic Lupus Erythematosus: Balancing Immunoregulation and Autoimmunity"

_ijms, 2025, doi:10.3390/ijms26073290_

Round 1

Reviewer 1 Report

Comments and Suggestions for Authors

I have reviewed the article titled "IL-10 in Systemic Lupus Erythematosus: Balance between Immunoregulation and Autoimmunity." This is a relevant study in which the authors determine interleukin-10 in patients with systemic lupus erythematosus and look for its association with different manifestations of the disease. It is a very interesting study. However, there are certain points that, in my opinion, need to be addressed:

  1. In the introduction, in the last paragraph, the authors talk about a review; however, the work they present is not a review.
  2. ¿Is IL-10 influenced by estrogen?
  3. Since so few patients were male, how did you assess the influence of hormones on the analyses? Wouldn't it have been better to evaluate men and women separately?
  4. What is the purpose or importance of obtaining a p-value for gender? From my point of view, this is not important. However, if the authors justify it, it might be necessary to retain it. If not, I suggest removing it from Table 3.
  5. How many times did the authors measure serum IL-10?
  6. Were the results consistent across measurements?
  7. How many male patients had IL-10 concentrations greater than 200 pg/ml? If only one, then this value is sufficient to generate a large dispersion of IL-10 values, which is not indicative of men having high IL-10 concentrations. Furthermore, the very small sample size of male patients means that any atypical data greatly influences the final mean. What can the authors say about this?
  8. What controls did the authors use (perhaps people without the disease) to say that IL-10 is increased?
  9. Why didn't the authors choose only patients with active disease?
  10. From my perspective, the results are inconclusive, since, as the authors themselves mention, the sample size is very low, especially for the male group. I suggest the authors analyze both the male and female groups separately.

Author Response

The team that made this material thanks you for such a thorough evaluation and relevant comments.

Reviewer 2 Report

Comments and Suggestions for Authors

This paper has the merit to deal with an interesting topic. However, it is a mix of an original and a review paper, which makes it too long and difficult to read.

Authors should decide what they want to do.

In my opinion, Authors should write an original paper quoting the available knowledge  in Introduction, also underlining the lack of data that justify their work, and at the end in Discussion they should discuss  their data in the light of existing literature.

They also may want to write a separate review paper on the role of IL-10 in SLE but they cannot do the two things in the same paper.

As a whole, the quality of English is good and it would not be a big job deleting paragraph 2 as a whole and all the material to be used to write a review (including Fig 1 and 2 and tables 1 and 5) and leave the original material. Authors should also eliminate in results all the sentences commenting data which, if needed, should be reported in Discussion.

I also suggest to start Discussion with a brief summary of the main results before discussing them in the light of existing literature.

But, in general, Authors should try to shorten the paper and to be more concise.

Minor comments

page 7 line 229-230: do authors really mean one of the two (either...or) or both?

page 7, line 252: titers > 1.2??

Author Response

(The authors gave the same response as above.)

Round 2

Reviewer 1 Report

Comments and Suggestions for Authors

Although the authors corrected their manuscript, they did not send answer sheets for each of the questions and suggestions, so we have to search for the new lines and paragraphs they added. The manuscript, which is all in yellow, indicates that they corrected everything, but they did not take the time to indicate the corrections they made per question.

Comments on the Quality of English Language

I am not qualified to comment on the quality of English.

Author Response

We thank and appreciate the comments on this text, which will help us to accomplish a valuable work.

Although the authors corrected their manuscript, they did not send answer sheets for each of the questions and suggestions, so we have to search for the new lines and paragraphs they added. The manuscript, which is all in yellow, indicates that they corrected everything, but they did not take the time to indicate the corrections they made per question.

We truly appreciate the reviewer’s time and effort in re-evaluating our work. Initially, the revised manuscript was submitted with ”Track Changes” option activated, to allow identification of all modifications. However, it appears that during file processing, the entire text was marked in yellow highlighting. We regret the inconvenience caused by the lack of a detailed response. We have now prepared a comprehensive point-by-point reply addressing each suggestion, as requested. Thank you once again for your valuable feedback, which has helped us substantially improve the quality and clarity of the manuscript.

  1. In the introduction, in the last paragraph, the authors talk about a review; however, the work they present is not a review.

We thank the reviewer for pointing this out. We wanted to emphasize that the study includes a brief background summary of IL-10 in the context of SLE to support the rationale for the research. To avoid any confusion, we have revised the sentence accordingly by removing the expression referring to the work as a “review” and clarifying that the manuscript presents original research.

Specifically, we deleted the sentence that stated “A literature review was also conducted to contextualize findings within existing data” (abstract) as well as the phrase “In this review, we provide an overview of the role of serum IL-10 in SLE pathogenesis”

We have retained the following revised version: Given IL-10’s dual immunoregulatory and pro-inflammatory functions [7],  we aimed to investigate its clinical relevance in SLE. Despite growing interest in IL-10 as a biomarker, existing studies offer conflicting results regarding its correlation with disease activity and clinical features, and few address these associations in real-life, treatment-exposed cohorts. In this original study, we analyzed the correlation between serum IL-10 levels and specific autoantibodies, disease activity, and clinical manifestations including arthritis, serositis, skin involvement, renal, hematological, and neurological involvement in a cohort of SLE patients. To support the rationale for our investigation, we also briefly summarized the current understanding of IL-10's role inLE.

  1. Is IL-10 influenced by estrogen?
  • Existing evidence suggests that estrogen can influence IL-10 production, particularly in immune cells such as B cells and monocytes. In response, we have addressed this point in the Discussion section by including a paragraph that also considers the influence of other sex hormones, such as testosterone and progesterone, on IL-10 expression.

We’ve added the following paragraph:

Sex-based differences in immune function have been extensively documented. Women have been demonstrated to respond more protectively than men to immunologically stimulating events. The high prevalence of female sex bias among autoimmune disease patients supports the notion that women, while having stronger immune responses, may also be more susceptible to developing autoimmunity and losing self-tolerance [45]. In this context, sex hormones appear to influence IL-10 production through multiple mechanisms. Estrogen regulates IL-10 expression in peripheral blood mononuclear cells from SLE patients, while androgens such as testosterone enhance Th2 differentiation and IL-10 production by T cells, and also act directly on CD4⁺ T lymphocytes to increase IL-10 levels [46,47]. Interestingly, in contrast, progesterone and estradiol have been reported to inhibit IL-10 production by activated marginal zone B cells, suggesting a complex, context-dependent regulation of this cytokine [48]. In light of these observations from the literature, we also explored the relationship between gender and IL-10 levels in our cohort.

  1. Since so few patients were male, how did you assess the influence of hormones on the analyses? Wouldn't it have been better to evaluate men and women separately?
  • The number of male patients in our cohort was limited (N = 9), which restricted the ability to perform a robust, sex-stratified analysis or to draw firm conclusions regarding the hormonal influence on IL-10 levels. While we reported a statistically significant difference in IL-10 levels between males and females, we agree that this finding should be carefully interpreted and acknowledged as a limitation. We have clarified this in the Discussion section.

We’ve added the following paragraph:

In our study, we observed a significant correlation between IL-10 levels and gender, with higher IL-10 concentrations in male patients compared to females. However, the small number of male participants limits the strength of this observation. Therefore, the potential influence of sex hormones on IL-10 levels in SLE should be further investigated in larger, sex-balanced cohorts.

  1. What is the purpose or importance of obtaining a p-value for gender? From my point of view, this is not important. However, if the authors justify it, it might be necessary to retain it. If not, I suggest removing it from Table 3.
  • While gender was not a primary focus of the study, we observed a notable difference in IL-10 levels between males and females. We considered it relevant to report this finding. The p-value was included to statistically support the observed difference and to inform potential future investigations into sex-based immunological differences. However, we agree that this is not a central objective of our study and we acknowledge that it may distract from the main focus (which was to assess the correlation between IL-10 levels and disease activity, clinical manifestations, and immunological parameters in SLE). As such, we have removed the p-value for gender from Table 3.

Table 1. Association of IL-10 levels with demographic and clinical characteristics.

Parameter

Mean ± SD

p-value (Pearson)

Disease Duration (years)

10.38 ± 9.73

0.517

Age (years)

51.17 ± 15.37

0.333

SLEDAI Score

3.61 ± 3.24

0.398

  1. How many times did the authors measure serum IL-10?
  • Serum IL-10 was measured once for each patient, based on a single blood sample collected at the time of clinical evaluation. We have now clarified this in the Materials and Methods section by adding additional information.

We’ve added: Serum IL-10 concentrations were measured once for each patient at the time of enrollment using a commercially available enzyme-linked immunosorbent assay (ELISA) kit

  1. Were the results consistent across measurements?
  • Serum IL-10 was measured only once per patient at the time of enrollment and clinical evaluation; therefore, we did not assess intra-individual variability or consistency across multiple measurements. We acknowledge this as a limitation of the study and have clarified this issue in the revised manuscript by adding a paragraph addressing it in the Discussion section.

We’ve added the following paragraph:

Although our findings add to the current understanding of IL-10 in SLE, some limitations should be acknowledged. Serum IL-10 was measured only once per patient, which did not allow us to assess intra-individual variability or fluctuations over time.

  1. How many male patients had IL-10 concentrations greater than 200 pg/ml? If only one, then this value is sufficient to generate a large dispersion of IL-10 values, which is not indicative of men having high IL-10 concentrations. Furthermore, the very small sample size of male patients means that any atypical data greatly influences the final mean. What can the authors say about this?
  • Only one male patient in our cohort had an IL-10 concentration above 200 pg/mL (236.8 pg/mL), while the second highest value was 73.6 pg/mL. We agree that this extreme value contributes to the dispersion of IL-10 values among males. To address this, we have added a clarification in the manuscript noting this point and encouraging cautious interpretation of sex-based differences. This issue is now clearly stated in the Results and Discussion sections.

We’ve added the following paragraphs:

RESULTS: Notably, only one male patient had an IL-10 level exceeding 200 pg/mL (236.8 pg/mL), and this value contributed to the overall dispersion of IL-10 values within the male subgroup.

DISSCUTIONS: Moreover, one male patient had an extremely elevated IL-10 value, which may have disproportionately influenced group-level comparisons

  1. What controls did the authors use (perhaps people without the disease) to say that IL-10 is increased?
  • We thank the reviewer for raising this point. Our study did not include a healthy control group, as the primary objective was to assess correlations between IL-10 levels and clinical or immunological features within an established SLE cohort. The absence of a control group limits our ability to compare IL-10 levels with those of healthy individuals, thus we have acknowledged this limitation in the revised Discussion and have referred to prior studies demonstrating elevated IL-10 levels in SLE patients compared to healthy controls.

We’ve added the following paragraph: The study did not include a healthy control group, which limits the ability to determine whether IL-10 levels were elevated compared to baseline physiological levels.

Our manuscript also includes the following paragraph: A considerable number of human investigations have revealed elevated levels of IL-10 in the serum of SLE patients when compared to controls [3,22,24,26–44].

  1. Why didn't the authors choose only patients with active disease?
  • Our intention was to include a real-world SLE cohort encountered in practice, which means both active and inactive cases. The majority of our patients had inactive disease at the time of evaluation, likely due to ongoing immunosuppressive treatment. We agree that including only patients with active disease might have increased the likelihood of detecting certain correlations (particularly with disease activity scores), but we believe that this approach, that provides insight into IL-10 levels across different disease states, represents more better the treated SLE population commonly encountered in rheumatology clinics.
  1. From my perspective, the results are inconclusive, since, as the authors themselves mention, the sample size is very low, especially for the male group. I suggest the authors analyze both the male and female groups separately.
  • We acknowledge the reviewer’s concern. We acknowledge that the sample size, particularly in the male subgroup (N = 9), is small and limits the robustness of sex-stratified analyses. Given this limitation, we opted not to perform detailed statistical comparisons between males and females beyond descriptive analysis, as subgroup testing could lead to overinterpretation of results. We have clarified this in the revised manuscript and emphasized the need for future studies with larger population to better explore sex-specific IL-10 dynamics in SLE.

Future research should focus on larger, prospective, and sex-stratified cohorts to further elucidate the role of IL-10 in disease pathogenesis, activity, and as a potential biomarker in SLE.

Reviewer 2 Report

Comments and Suggestions for Authors

Authors have done a good job.

However, in different points of Results they forgot to delete comments on results

Anyway, I think the paper is still too long. I suggest to try to shorten Discussion at least of 30%

Specific comments

page 1, line 32: please replace "accentuate" with "underline"

page 2, lines 69-71 Please replace

Their involvement in SLE is significant, potentially serving as biomarkers to monitor disease progression.

with

Because their involvement in SLE is significant, they can potentially be used  as biomarkers to monitor disease progression.

page 2, lines 76-77

please replace

try to regulate and reduce inflammatory responses,

with

can down-regulate inflammatory responses,

page 2 lines 88-89

Please delete the sentence

To support the rationale for our investigation, we also briefly summarized the current understanding of IL-10's role in SLE.

page 2, line 98 "Applied" should be deleted

page 4, line 263

please replace

The study constituted 88 patients with SLE. The mean disease duration was 10.38 ± 9.73 years, varying between 0 and 37 years

with

The 88 patients with SLE, enrolled in the study, had a  mean disease duration of  10.38 ± 9.73 years, varying between 0 and 37 years

page 4, lines 281-284

The sentence

However, the small number of male partic- ipants (N = 9) limits the statistical power of this comparison. Consequently, this subgroup is considered methodologically inconsistent; therefore, the findings should be interpreted with caution.

should be deleted because is not a result but a comment of results

page 5, lines 313-316

While Pearson's correlation suggests a weak negative relationship, implying that IL-10 levels may slightly decrease as disease duration increases, the p-value indicates a lack of statistical significance.

To be deleted for the above mentioned reasons

page 7, line 386

providing key insight into its potential role in SLE.

To be deleted for the above mentioned reasons

page 7, lines 395-397

Spearman’s rank correlation, a non-parametric method, was preferred to assess asso- ciations between IL-10 and other cytokines. This methodological choice ensures the ro- bustness of our findings

Again this is a comment and should not be  in Results

Page 10, line 537

The high prevalence of female sex bias ......... What authors mean with "sex bias"?

Page 12, lines 676-677

Our findings were interpreted in the light of existing literature to emphasize the role of IL-10 in SLE.

 This sentence sounds strange

Could you please rephrase?

Author Response

We thank and appreciate the comments on this text, which will help us to accomplish a valuable work.

Dear Reviewer,
We would like to sincerely thank you for your valuable guidance throughout this review process. Your initial recommendation “Authors should decide what they want to do” challenged us to reflect on the structure and purpose of our work.

As a result, we understood the importance of maintaining a clear distinction between a review and an original article. Although both types of contributions are valuable, each serves a different purpose and deserves its own space. We now recognize how separating them helps improve clarity and makes the content more engaging for readers.

Your critical feedback motivated us to revise our manuscript substantially, and we are truly grateful for the constructive direction you provided. It has not only improved this paper but has also taught us important lessons we will carry forward in our future research.

Authors have done a good job.

  • However, in different points of Results they forgot to delete comments on results

We have re-viewed the Results section and removed any remaining comments. We also confirm that we have thoroughly addressed all additional points raised in the specific comments below, and we sincerely appreciate your constructive and detailed feedback throughout the manuscript. Your suggestions have been extremely valuable.

  • Anyway, I think the paper is still too long. I suggest to try to shorten Discussion at least of 30%

We have re-read the Discussion section and thoroughly revised it, removing redundant or over detailed phrases that made the text unnecessarily long or difficult to follow. As a result, the section has been shortened. We hope that the revised version improves the clarity and overall readability of the manuscript.

Specific comments

  • page 1, line 32: please replace "accentuate" with "underline"

page 2, lines 69-71 Please replace

  • Their involvement in SLE is significant, potentially serving as biomarkers to monitor disease progression.

with

Because their involvement in SLE is significant, they can potentially be used  as biomarkers to monitor disease progression.

  • page 2, lines 76-77

please replace

try to regulate and reduce inflammatory responses,

with

can down-regulate inflammatory responses,

  • page 2 lines 88-89

Please delete the sentence

To support the rationale for our investigation, we also briefly summarized the current understanding of IL-10's role in SLE.

  • page 2, line 98 "Applied" should be deleted

  • page 4, line 263

please replace

The study constituted 88 patients with SLE. The mean disease duration was 10.38 ± 9.73 years, varying between 0 and 37 years

with

The 88 patients with SLE, enrolled in the study, had a  mean disease duration of  10.38 ± 9.73 years, varying between 0 and 37 years

  • page 4, lines 281-284

The sentence

However, the small number of male partic- ipants (N = 9) limits the statistical power of this comparison. Consequently, this subgroup is considered methodologically inconsistent; therefore, the findings should be interpreted with caution.

should be deleted because is not a result but a comment of results

  • page 5, lines 313-316

While Pearson's correlation suggests a weak negative relationship, implying that IL-10 levels may slightly decrease as disease duration increases, the p-value indicates a lack of statistical significance.

To be deleted for the above mentioned reasons

  • page 7, line 386

providing key insight into its potential role in SLE.

To be deleted for the above mentioned reasons

  • page 7, lines 395-397

Spearman’s rank correlation, a non-parametric method, was preferred to assess asso- ciations between IL-10 and other cytokines. This methodological choice ensures the ro- bustness of our findings

Again this is a comment and should not be  in Results

  • Page 10, line 537

The high prevalence of female sex bias ......... What authors mean with "sex bias"?

We thank the reviewer for pointing this out. By “female sex bias”, we refer to the well-documented predominance of autoimmune diseases, including SLE, in female patients compared to males. To avoid any ambiguity, we have revised the phrasing in the manuscript to clarify this meaning and replaced “female sex bias” with “female predominance in autoimmune diseases”, which more accurately reflects the intended message.

  • Page 12, lines 676-677

Our findings were interpreted in the light of existing literature to emphasize the role of IL-10 in SLE.

 This sentence sounds strange

Could you please rephrase?

To highlight the role of this cytokine, we interpreted our findings in relation to previously published studies on IL-10 in SLE.

  • We noticed that the criteria “Are the conclusions supported by the results?” was marked as “Must be improved”

We took this as feedback and we have revised the Conclusions section. We hope that the new version

offers a more accurate synthesis of our study. We are grateful for the time and attention you have dedicated to our work.

Round 3

Reviewer 2 Report

Comments and Suggestions for Authors

All my suggestions and criticisms were well addressed

No further comments